# Bridging the Gap between Medical Tabular Data and NLP Predictive Models: A Fuzzy-Logic-Based Textualization Approach

Chérubin Mugisha [†] [ID] and Incheon Paik *,[†]

Department of Computer Science and Engineering, University of Aizu, Aizu-Wakamatsu 965-8580, Japan
* Correspondence: paikic@u-aizu.ac.jp; Tel.: +81-242-37-2796
† These authors contributed equally to this work.

**Abstract:** The increasing use of electronic health records (EHRs) generates a vast amount of data, which can be leveraged for predictive modeling and improving patient outcomes. However, EHR data are typically mixtures of structured and unstructured data, which presents two major challenges. While several studies have focused on using machine learning models to predict patient outcomes, these models often require data to be in a structured format, which may lead to the loss of important information. On the other hand, unstructured data, such as narrative reports, can be noisy and challenging for natural language processing applications and interoperability. Therefore, there is a need to bridge the gap between structured EHR data and NLP-based predictive models. In this paper, we propose a fuzzy-logic-based pipeline that generates medical narratives from structured EHR data and evaluates its performance in predicting patient outcomes. The pipeline includes a feature selection operation and a reasoning and inference function that generates medical narratives. We then extensively evaluate the generated narratives using transformer-based NLP models for a patient-outcome-prediction task. We furthermore assess the interpretability of the generated text using Shapley values. Our approach has demonstrated comparable performance to the benchmark baseline models with an F1-score of 93.7%, while exhibiting slightly improved results in terms of recall. The model demonstrated proficiency in the preservation of information and interpretability inherited from nuanced and structured narratives. To the best of our knowledge, this is the first study to demonstrate the ability to transform tabular data into text to apply NLP for a prediction task.

**Keywords:** fuzzy logic systems; defuzzification; medical data; NLP; outcome prediction; transformer-based model

## 1. Introduction

In recent years, medical data have been increasingly being generated at an unprecedented rate, leading to a vast amount of information that can be utilized to improve healthcare outcomes. Electronic health records (EHRs) have been identified as one of the most promising sources of data in healthcare for research and clinical practice [1]. However, EHR data are often heterogeneous. These well-documented structured data (demographic information, diagnosis, laboratory test results, monitoring data) and unstructured data (clinical notes) are stored in various formats [2]. The large amount of data in EHRs makes manual analysis and interpretation a challenging and time-consuming task, leading to automated methods for processing these data.

Machine learning algorithms have been proposed as a powerful tool for processing EHR data for various tasks, including outcome prediction, patient-risk stratification, and disease diagnosis [3]. However, using machine learning models to perform these tasks has two significant challenges. On the one hand, the characteristics of structured data may vary from one patient to another depending on their diseases and periods of stay. Additionally, structured medical data require feature selection as a data regularization

mechanism for standard models, such as neural networks, tree-based models, and other mainstream modeling methods [4]. This normalization process cuts out various features and drastically reduces the data samples. On the other hand, unstructured data, such as narrative reports, can be noisy and challenging for natural language processing (NLP) applications and interpretability. On the other hand, although NLP models have shown impressive performance in information extraction and data representation, EHR clinical narratives and reports are particularly challenging for NLP applications such as outcome prediction and interpretability. This is partly because applying NLP to such data requires specific standards, since document representation relies on dictionaries and vocabularies from common natural languages [5].

To address these challenges, we propose a fuzzy logic (FL) based pipeline that generates medical narratives from structured data. This approach bridges the gap between the diverse and predominant medical tabular data and NLP predictive models, enabling the processing of rich structured data with the power of NLP models. Through this research, we propose a rule-based pipeline to describe a patient using structured data by generating a text document and evaluating the usefulness of the synthetic summary by transformer-based models to predict a patient's outcome.

The first step consists of feature selection inspired by a baseline study from the literature review [6], then from the selected features. We divide the data extraction task into small clinical-services-related data clusters. We then textualize the features using preconceived prompts according to the availability of the feature values. Finally, we evaluated the effectiveness of the generated text in a downstream text classification task using several transformer-based NLP models, including an optimized RoBERTa-based model [7], BERT [8], and a pre-trained biomedical language representation model (BioBERT) [9].

The contribution of this research is summarized as follows:

- We are proposing an approach to generating clean and comprehensive medical narratives to describe a patient through a textualization process of the medical tabular data.
- A superficial application of fuzzy theory through defuzzification to create a syntax dictionary for substituting the numerical values of medical parameters. This textualization preserves the uncertainty and vagueness inherent in medical data while still allowing for the application of NLP methods.
- A comprehensive study utilizing the data generated to solve a patient-outcome-prediction problem based on NLP classifier models with respect to their ability to "explain" their predictions.

To the best of our knowledge, no prior research has been conducted using an FL-based textualization approach to bridge the gap between medical tabular data and natural language processing (NLP).

The rest of the manuscript is organized as follows. Section 2 highlights related publications and preliminary concepts. Section 3 presents our methods and approach for data transformation and modeling. Section 4 presents the evaluation of our results in a prediction task. Section 5 concludes the paper and outlines the limitations of our methods, and suggests future work.

## 2. Background

Biomedical data mining aims to extract knowledge from large amounts of biomedical data. The goal of this process is to identify and understand patterns and relationships within the data that can be exploited later to improve healthcare and understand outcomes. With machine learning, biomedical data mining requires a data transformation that involves converting raw data into a format that can be easily manipulated with the available tools for better performance [10]. Various normalization techniques include:

- Standardization, which scales data to a common range.
- Normalization, which scales data to a common distribution.
- Discretization, which converts continuous data into discrete data.

Data discretization can be performed by binning, which groups data into a specified number of bins, or by clustering data based on similarity. Discretization strives to improve the interpretability of biomedical data. For EHR data, these methods can be computationally expensive but can also lead to a massive loss of information.

In recent years, many studies have proposed various techniques to process and analyze medical data. For instance, deep learning models have been used to predict clinical outcomes, such as patient mortality, length of stay, and readmission rates, using electronic health records (EHRs) [11,12]. A study by Choi et al. [3] proposed a recurrent neural network (RNN) model that uses clinical notes to predict hospital readmission. Their approach proposes an interpretable predictive model for healthcare that uses a reverse time attention mechanism to capture relevant information from the patient's historical medical records. Similarly, a study by Purushotham et al. (2018) proposed a deep learning model that incorporates both structured and unstructured data from EHRs to predict patient mortality [12]. In data transformation, several works have been presented. For instance, Arnaud et al. [13] proposed a distillation method to extract structured data from unstructured text.

However, few studies have suggested transforming structured data into unstructured free text. Structured data naturally encourage accuracy among machine learning models, and interoperability, but NLPs are still black boxes. However, processing unstructured data, such as clinical notes, can be challenging due to the variability and complexity of clinical language [14].

In their book, Jang et al. [15] proposed a comprehensive theory on applying fuzzy logic and machine learning to address the uncertainty in a data transformation while emphasizing the interpretability of the result. This work inspireed us to solve the vagueness inherent in medical data.

### 2.1. Fuzzy Theory

Traditionally, FL is a science that makes machines think and understand how humans do by proposing fuzzy sets to manage imprecise and vague knowledge [16,17]. As computational intelligence techniques, fuzzy methods are used for effective decision making to bridge the gap between human and machine intelligence by resolving the ambiguity of terms. The paradigm of computing with words was a rational consequence of fuzzy theory reasoning for computers [18]. However, FL, with its concept of the linguistic variable and application to approximate reasoning, is a method of computing with words [19]. While today's technologies can only simulate that computation, we still cannot compute with words as long as the encoding process transforms words back into numbers. Therefore, this approach can be assimilated into a rule-based algorithm to define numerical variables with words. While numbers are used in statistical and machine learning models, humans understand better in natural language. Therefore, using NLP should require a data transformation of the numerical values into more meaningful terms for such models.

### 2.2. Hybrid Fuzzy-Based Models for Text Generation

Hybrid modeling integrating deep neural networks (DNN) and fuzzy systems has been defined in various ways for diverse reasons [20,21]. One of the main motivations for that symbiosis is DNN optimization [22,23]. As an illustration of how FL is used in NLP is in natural language understanding (NLU). Fuzzy logic can be used to interpret the meaning of a natural language input by taking into account the context and the degree of uncertainty of the input [24]. For example, a statement such as "The patient has a high blood pressure..." could be interpreted differently. FL can be used to determine the degrees of membership of the input to different categories, such as "*normal*", "*elevated*", "*High*", or "*Hypertensive*", to allow a more accurate interpretation of the input based on that information.

Another example of the use of FL in NLP is in natural language generation (NLG). FL can be used to generate natural language output that is more human-like and less rigid than traditional rule-based systems by taking into account context and degree of

certainty [18,25]. Let us assume a set of linguistic variables that represent different features or attributes of the text:

$$X = x_1, x_2, ..., x_n \qquad (1)$$

and a set of fuzzy sets that represent the values of the linguistic variables.

$$A = A_1, A_2, ..., A_n \qquad (2)$$

The membership functions of the fuzzy sets are used to represent the degree of membership of each value in a linguistic variable.

FL can be utilized to generate text by using a fuzzy inference system, which consists of a set of rules that define the relationships between the linguistic variables. The rules can be defined as *IF $x_1$ is $A_1$ AND $x_2$ is $A_2$ THEN $x_3$ is $A_3$*. The rules are used to emanate a set of fuzzy output variables that are fused, and a reverse engineering mechanism (defuzzification) is applied to generate the final text. This fuzzy text generation can be expressed as:

$$y = \sum_{i=1}^{n} (w_i * \mu A(x_i)) \qquad (3)$$

where $y$ is the output text, $w$ is the weight of each rule, and $\mu A(x)$ is the membership function of the fuzzy set for each linguistic variable.

### 2.3. Defuzzification

The fuzzy membership degrees are used to define a crisp output or a single, definite-meaning representation of the input [26,27]. This reverse engineering mechanism has three main methods:

- Centroid method: It calculates the center of mass of the fuzzy set, which describes the average value of the set.

$$x_{centroid} = \frac{\sum_{i=1}^{n} x_i * \mu A(x_i)}{\sum_{i=1}^{n} \mu A(x_i)} \qquad (4)$$

where $x_{centroid}$ is the crisp value resulting from defuzzification, $x_i$ is a sample value, and $\mu A(x_i)$ is the membership degree of $x_i$ in fuzzy set $A$.

- Maximum membership degree method: This method specifies the value with the highest membership degree as the crisp output.

$$x_{max} = \arg \max_{x} \mu A(x) \qquad (5)$$

where $x_{max}$ is the crisp value resulting from defuzzification and $\arg \max_x$ is the argument that maximizes the membership function.

- Mean of maximum (MOM) method: The MOM method calculates the average value of the values that have maximum membership degrees.

$$x_{MOM} = \frac{\sum_{i=1}^{n} x_i * [\mu A(x_i) = \mu_{max}]}{\sum_{i=1}^{n} [\mu A(x_i) = \mu_{max}]} \qquad (6)$$

where $x_{MOM}$ is the crisp value resulting from defuzzification, $x_i$ is a sample value, $\mu A(x_i)$ is the membership degree of $x_i$ in fuzzy set A, $\mu_{max}$ is the maximum membership degree in fuzzy set A, and $[\mu A(x_i) = \mu_{max}]$ is a binary variable equal to one if $\mu A(x_i) = \mu_{max}$, and it is equal to zero otherwise.

With this research, we converge on this traditional use of fuzzy logic theory in NLU and NLG. We propose a way of using balanced linguistic theory and clinical features occurring in a tabular format to build comprehensive patient descriptive documents using defuzzification methods. In the following section, we describe our approach to the

construction of the fuzzy set and how our evaluation yielded the best performance in a patient-outcome-prediction task.

## 3. Materials and Methods

### 3.1. Introduction

The ultimate goal of textualizing tabular data is to propose a predictive model based on a general understanding of a patient's status. The best part of this is the use of as much information as available from the EHR, without compromising on using certain parameters in the process of handling missing data and outlier values. Numerical models require the regularity of the input features. However, medical data are full of such irregularities that an extensive data processing step is needed, including sample selection, data balancing, and normalization. Ideally, a patient's complete medical description should:

- Include the patient's name, a unique identifier, and the location of his or her hospitalization.
- Reflect the continuum of patient care in a chronological order.
- Contain data recorded on admission, handover, and discharge.
- Be dated and signed by their author.

In most of the publicly available medical datasets, these important data are missing due to the de-identification process. This will not make any exceptions for the synthetic data when describing a patient.

In EHR, an equivalent description can mostly be found in clinical reports, which usually use natural language to describe a patient. However, free narratives are irregular and hard to process due to conventional writing, which can vary from one health center to another. Our objective is to describe a patient in a natural way using medical data, mimicking real-world datasets. Moreover, we want to bring the benefits of NLP and transformers to more use cases in medical predictive models while avoiding the pre-processing step required by the EHR narratives and comparing our results with the existing tabular-based models.

### 3.2. Data Acquisition and Mining

Medical Information Mart for Intensive Care-III (MIMIC-III) is a publicly available dataset with real medical data from over 38,597 distinct patients admitted to an intensive care unit (ICU) [28]. The data are distributed as CSV files that can be imported and mapped to a relational database such as MySQL. Using SQL queries, datasets were extracted and processed in a Python notebook. In order to benchmark the effectiveness of our proposed method later, we utilized the same data-inclusion criteria as our baseline model from the literature [6]. We selected patients admitted or transferred to the Cardiac Surgery Recovery Unit (CSRU), Medical ICU (MICU), and Surgical ICU (SICU), and used "emergency or Urgent" as ADMISSION_TYPE. Please refer to the mentioned paper for details on the inclusion criteria and statistics. To keep the relations between entities for the next step, we queried the database in five distinct dataset clusters containing administrative information, diagnosis related information, laboratory tests, vital signs, and procedures. Figure 1 shows a summary of the process from the data extraction down to the next generation.

A common problem for any medical-outcome-prediction study is class imbalance. Within our dataset, few patients died during their ICU hospitalization—a minority of 5058 among 37,111 unique admissions. This shows a fatality rate of 13.62% in our population. Different techniques for handling imbalanced data exist; for our case, in order to keep the integrity of the data, downsampling the majority class by random selection was utilized. However, this technique has the consequence of cutting out some potential knowledge from the majority class. To limit this information loss, we sampled the new dataset to 40% for the fatality class and 60% for the discharged class.

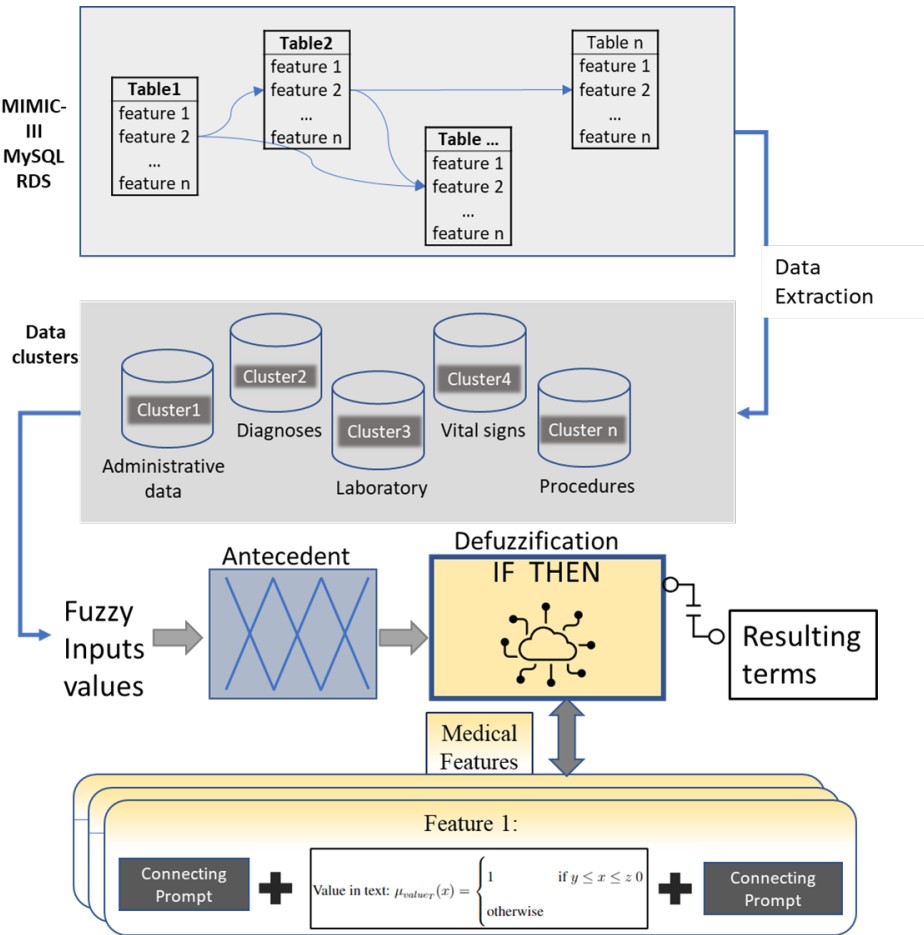

**Figure 1.** Summary of the data extraction and synthetic narratives of the generation pipeline.

### 3.3. Proposed Model: Data Textualization

The literature has shown the most relevant features to predicting a patient's outcome in an ICU [6,11]. However, most of the authors were also limited by constant missing data through the population sets and the targeted case of study. In our case, our limit could be determined by the NLP model itself for not performing well with numerical data. As described in the following section, a rule-based algorithm based on FL theory can be used to map numerical values with classes that can be understood by a modern LM. Generating synthetic text data with fewer numbers and more key terms allows us to build a more comprehensive NLP model to accomplish a task such as classification.

To describe a patient inherently, we conducted the generation of narratives with the help of key phrases. These key phrases connect medical parameters extracted from different EHR tables to ensure the semantic and syntactic integrity and the relevance of the generated text.

### 3.4. Feature Engineering

One of the most typically used methods in FL is the membership function, which assigns a degree of membership to each element of the input set, based on its resemblance to the set or category in question.

The MIMIC-III dataset uses the International Classification of Diseases in its ninth version (ICD9) to encode and classify diagnoses. These codes are the primary sources of information related to each patient's main complaint, comorbidities, and phenotypes. For our text generation process, we extracted all the codes related to each patient of our population found in the ADMISSIONS and DIAGNOSES_ICD tables. To map the codes with their label, we used a python library "*icd9cms*" that takes the ICD9 codes as an

input, and we specifically output the most granular label from the hierarchy of the ICD9 nomenclature tree [29]. This step provided us with the clinical name associated with each ICD9 code, which we appended to other prompts and text related to patient identification, procedures, vitals, and laboratory tests and results.

In the following section, we describe our methods of transforming all those numerical data into text using a defuzzification system.

### 3.5. Defuzzification System Architecture

Our ultimate goal is to evaluate our approach with a language model. However, language models understand better textual context than numerical context. For instance, a patient's blood pressure being annotated as "*140/101* mmHg" may not provide much meaning for a language model. Therefore, its interpretation in medical terms (hypertension, specifically, stage 2 hypertension) has more potential to be well understood by a LM.

The fuzzy theory defines the linguistic variable by:

$$(X, T(x), U, G, M) \tag{7}$$

$X$ is the variable, $T(x)$ is the set of terms, $U$ is the universe of discourse, $G$ represents the syntax rules, and $M$ defines the semantic rules. For our case, $X$ represents the medical variables, $T(x)$ are the clinical interpretations, and $U$ groups the values of each medical parameter [19].

A defuzzification dictionary for blood pressure readings could be mapped to categories such as "*Low*", "*Normal*", and "*High*". In our algorithm, instead of computing the centroid of the fuzzy output, we simply compute the maximum degree of membership among all categories using a binary rule approach. This approach can be more efficient and easier to implement, but it may not be as accurate as the centroid approach in certain cases.

In Algorithm 1, the membership function $\mu_{Normal}(x)$ returns a value of one if the blood pressure reading $x$ falls within the range of 90 to 139 mmHg, indicating that the reading is "*Normal*". The value of $\mu_{Normal}(x)$ is 0 for readings outside of this range. Similarly, membership functions can be defined for each of the other categories which we intend to substitute with words in the textualization process. However, the encoding part of the logic can be handled by a multidimensional LM to vectorize these entities of words. As in fuzzy theory, where each linguistic variable is described by a "set of terms", to textualize our medical features, each feature's value is represented by a term instead of a number. Our approach utilizes binary discrimination to allocate a category to each value [30]. Table 1 reports the set of terms with the range and source of reference for each parameter.

---

**Algorithm 1.** Defuzzification for blood-pressure categories.

---

Fuzzy blood pressure reading $x$ Blood pressure category Compute the degree of membership of $x$ in each category using fuzzy sets or rules, such as:

- Low: $\mu_{Low}(x) = \begin{cases} 1, & \text{if } x \leq 90 \text{ mmHg} \\ 0, & \text{otherwise} \end{cases}$

- Normal: $\mu_{Normal}(x) = \begin{cases} 1, & \text{if } 90 \text{ mmHg} < x \leq 139 \text{ mmHg} \\ 0, & \text{otherwise} \end{cases}$

- High: $\mu_{High}(x) = \begin{cases} 1, & \text{if } x > 139 \text{ mmHg} \\ 0, & \text{otherwise} \end{cases}$

Compute the maximum degree of membership among all categories, such that:
$$\mu_{max} = \max \mu_{Low}(x), \mu_{Normal}(x), \mu_{High}(x)$$
If $\mu_{max} = \mu_{Low}(x)$, return "Low" as the blood pressure category.
If $\mu_{max} = \mu_{Normal}(x)$, return "Normal" as the blood pressure category.
If $\mu_{max} = \mu_{High}(x)$, return "High" as the blood pressure category.

---

**Table 1.** Medical parameters, set of category terms, and their ranges.

| Parameter | Range | Category | Reference |
|---|---|---|---|
| Age | 15–40<br>41–60<br>61–89<br>90+ | Young adult<br>Middle-aged adult<br>Old-aged adult<br>Very old-aged adult | [31] |
| Arterial Blood Pressure | Sys <90 mmHg<br>Sys 90–139 mmHg<br>Sys >139 mmHg | Low<br>Normal<br>High | [32] |
| Heart rate(HR) | <60 BPM<br>60–100 BPM<br>>100 BPM | Low<br>Normal<br>High | [32] |
| SpO2 | <92% BPM<br>>92% BPM | Low<br>Normal | [33] |
| Heart Rate (HR) | <60 BPM<br>60–100 BPM<br>>100 BPM | Low<br>Normal<br>High | [32] |
| Respiratory Rate | <12 BPM<br>12–25 BPM<br>>25 BPM | Low<br>Normal<br>High | [32] |

### 3.6. Machine Learning Models

In NLP, transformer-based models [34] have become a reference as the state-of-the-art in several natural language understanding tasks. In this research, we decided to use this representation over the fuzzification, since it captures relations between neighboring and distant words, whereas the Fuzzy encoder only considers one single word as an independent entity.

#### 3.6.1. BERT

The bidirectional encoder representations from transformers (BERT) model is a highly bidirectional, unsupervised language representation method pre-trained on an unlabeled plain text corpus from books and the English Wikipedia. The original model was presented in two versions, the $BERT_{BASE}$ with 12 encoders and 12 self-attention heads and $BERT_{LARGE}$ with 24 encoders and 16 bidirectional self-attention heads. We omit more details on the architecture, as it is well described in [8]. BERT utilizes the transformer encoder architecture based on a self-attention mechanism to represent a sequence of words or tokens in a higher dimensional space. We utilized the $BERT_{BASE}$ version, since our inputs have an average of 353 tokens.

#### 3.6.2. BioBERT

The biomedical language representation model for biomedical text mining (BioBERT) is a domain-specific language model [9]. This baseline model initialized its weights from BERT and uses PubMed abstracts and PMC full-text articles to fine-tune its understanding of the medical domain. Please, refer to the original paper for more details on the training process and the performance of the resulting model.

During the tokenization process, two additional tokens are used: a [CLS] token as an input starter and [SEP] to mark the end of the input sequence. Thus, a sequence $S$ for these models is represented by $[cls, t1, ..., tn, sep]$, where $t$ is a word or a subword of $S$. The maximum length of the input sequence is 512 tokens. The goal of using tokens is to represent any words and avoid OOV words. However, BERT and BioBERT are token based-models; thus, some words will be broken down into characters if such entities are not present in the 30,000-token vocabulary files of those models.

### 3.6.3. BioBERTa

BioBERTa is a pre-trained RoBERTa-based language model designed specifically for the biomedical domain [7]. Like other domain-specific LMs, BioBERTa has been trained on a diverse range of biomedical texts—mostly electronic health records and raw medical notes—to learn the language patterns, terminology, jargon, and knowledge relevant to the biomedical domain. BioBERTa was optimized in the pretraining process by adopting the modifications of the source model, such as dynamic masked language modeling, no next-sentence prediction task, and most importantly, a WordPiece tokenizer that suppresses the out-of-vocabulary (OOV) occurrences. This model demonstrated high performance on several named-entity recognition tasks and showed the best fertility rate for biomedical texts.

To fine-tune these three models for a classification task, we appended a classification layer on top of the last hidden layer with a given loss function, and this can be performed on the output of the [CLS] token alone. For our case, we utilized the [CLS] token and a logistic regression classifier. We performed a hyperparameter search to find the best set of training epochs, learning rate, and batch size that optimizes the result [35].

## 4. Results

The process of generating data begins with the extraction of features from the main MIMIC-III dataset. The extracted features were then individually merged in a fusion process to form a more comprehensive representation of the patient. In this operation, the features were contextualized using key phrases to semantically link them, thereby creating a coherent representation of a narrative. To ensure the quality of the generated data, a grammatical assessment was carried out to eliminate any unnecessary duplication of features or syntax errors that may have been introduced during the fusion process. This grammatical assessment helped to improve the coherence and consistency of the generated data and provided a better sequence of the features extracted from the main MIMIC-III dataset.

### 4.1. Generated Data

In order to provide a comprehensive understanding of each patient's hospitalization, we generated narratives for each of the 37,110 admission IDs in the dataset. The length of these generated texts was determined by the number of parameters each patient had, resulting in a dataset that includes both admission IDs and labels indicating the outcome of the patient's hospitalization.

When preparing the generated dataset for use with classification models, it was essential to ensure that it would fit within the limitations of the models. Using the BERT tokenizer, we counted the tokens of each input sentence, and the results were 1686 and 67, respectively, for the longest and shortest sentences. The median was 258 tokens. With this in mind, we excluded the normal values of the parameters from the training data. The rationale for this was that the primary purpose of medical procedures is to identify or treat abnormalities. Figure 2 shows the variation in the narrative's lengths before and after this step.

By keeping the normal values of the parameters, more than 2700 narratives have over the 512-token limit of our classification, and only fewer than 1600 will hit that limit without normal values.

### 4.2. Classification Results

To train and evaluate our two models, we used 10,116 input sentences and tested their performances for 2529 narratives. To ensure compatibility, we utilized the BERT-based, uncased tokenizer as BERT and BioBERT's tokenizer and the vocabulary that came with the pre-trained BioBERT files. BioBERTa has a custom byte-pair encoding (BPE) tokenizer of 50,265 tokens.

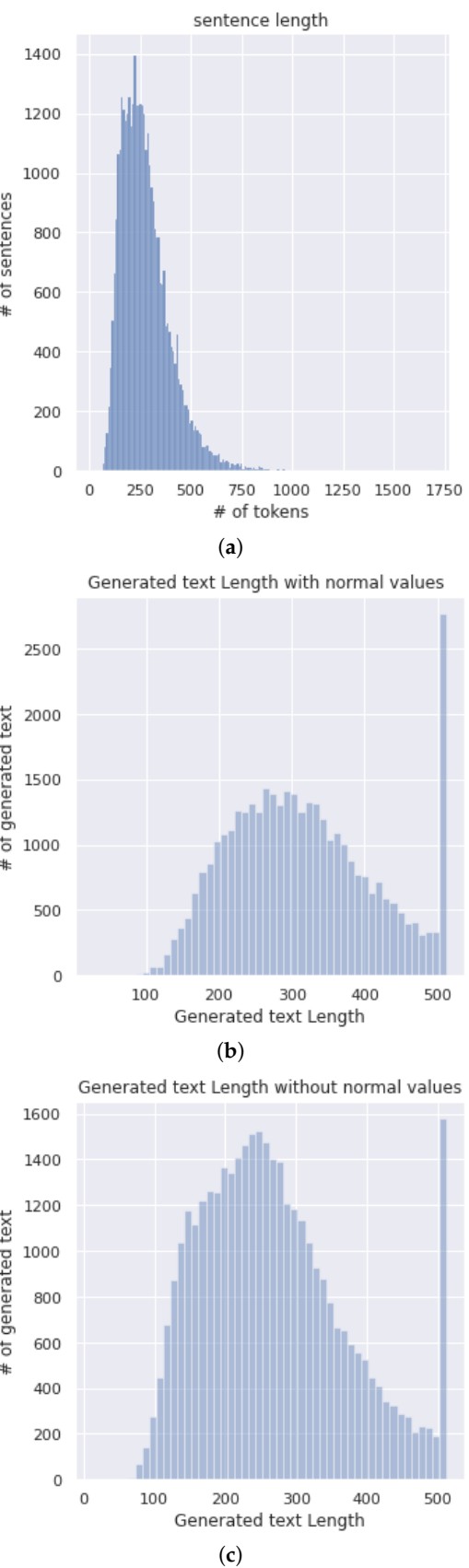

**Figure 2.** Generated narratives' lengths: These three graphs provide an overview of the lengths of our synthetic texts. (**a**) Overview of the generated text; (**b**) sentences of ≤512 tokens with normal values; (**c**) sentences of ≤512 tokens without normal values.

### 4.2.1. Input-Length-Variation Study

To understand the behavior and determine the optimal input size for each model, we conducted experiments using different input lengths of 512, 350, and 255 tokens. This allowed us to determine the most effective input size to achieve the best results. Preliminary results revealed that the best performance was achieved using a maximum input length of our models (512 tokens). For the rest of the experiment, we used tokenized inputs of a maximum length of 512 tokens for the three models.

### 4.2.2. Optimization of Hyperparameters

Hyperparameter optimization in NLP consists of selecting the optimal values for the model's hyperparameters to achieve the best performance in a downstream task by effectively capturing the patterns in the data and avoiding overfitting or underfitting. These hyperparameters define the configuration of the model, such as the learning rate, the batch size, and the number of hidden layers. For our case, we focused our attention on the training batch size, the learning rate, and the training epochs.

To develop an adaptive (sequential) hyperparameter search, we utilized a random search algorithm to erratically select different combinations in the provided ranges [36]. We focused our attention on the training batch size, the learning rate, and the training epochs. Figure 3a shows that with a batch size of 32, four training epochs, and a learning rate of $3.86 \times 10^{-5}$, can provide an F1-score of 93.47%.

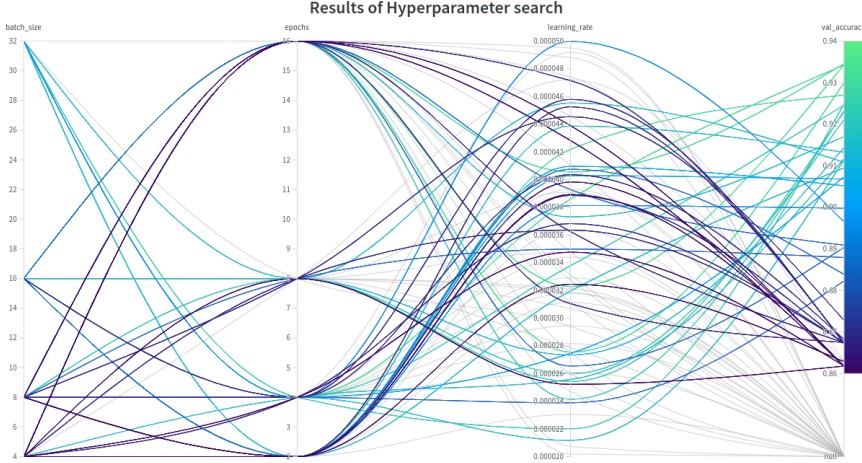

(**a**) Hyperparameter search in a predetermined range of values.

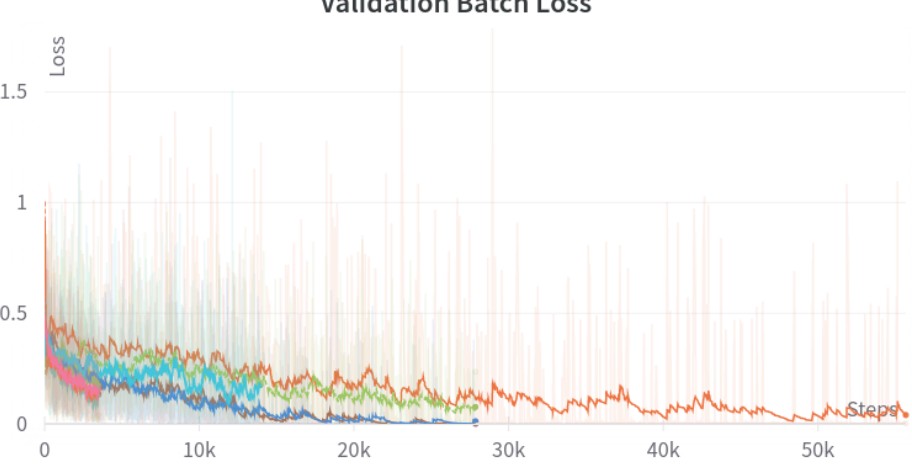

(**b**) Loss for the different model configurations

**Figure 3.** Hyperparameter search and validation loss.

### 4.2.3. Outcome-Prediction Results

Our test was to use the generated data and evaluate their importance in solving the problem of hospitalization outcomes. Our data were labeled as "0" if the patient was discharged and "1" if he died during his hospitalization.

We measured the efficiency of the in-hospital predictive models by the evaluation metrics, F1-score, precision, and recall between the fatality and survivor classes. Each reported score in Table 2 is an average of five experiments on BERT, BioBERT, or BioBERTa.

**Table 2.** Outcome prediction results from three different NLP models and a tabular data-based stacking model as a baseline.

| Model | Input Length | F1 | P | R |
|---|---|---|---|---|
| Stacking Model [6] | - | 0.937 | 0.964 | 0.911 |
| BERT | L = 256 | 0.849 | 0.815 | 0.886 |
| | L = 360 | 0.848 | 0.825 | 0.873 |
| | L = 512 | 0.858 | 0.847 | 0.870 |
| | L = 512 (optimized) | 0.897 | 0.887 | 0.895 |
| BioBERT | L = 256 | 0.851 | 0.817 | 0.887 |
| | L = 360 | 0.860 | 0.865 | 0.855 |
| | L = 512 (optimized) | 0.925 | 0.931 | 0.926 |
| BioBERTa | L = 256 | 0.854 | 0.797 | 0.921 |
| | L = 360 | 0.860 | 0.845 | 0.875 |
| | L = 512 | 0.879 | 0.849 | 0.891 |
| | L = 512 (optimized) | 0.937 | 0.94 | 0.931 |

The baseline paper [6] explored two different methods for approaching the task at hand. The first method involves the utilization of a list of unimodal baseline classifiers, including k-nearest neighbor (KNN), multilayer perceptron (MLP), linear discriminate analysis (LDA), logistic regression (LR), and decision tree (DT), applied to various experimental feature sets. The second method involves ensemble models, such as random forest, voting, bagging, and boosting, to improve the performance of the best single models. These evaluations were conducted both with and without a feature-selection step.

Building upon these two approaches, the paper introduced a stacking classifier algorithm based on the generalization stacking ensemble model, using LR as the metaclassifier. This stacking technique demonstrated impressive accuracy, with an F1-score, precision, recall, and AUC of 0.937, 0.964, 0.911, and 0.933, respectively.

The results displayed in Table 2 demonstrate the highly competitive performance of both models. BioBERTa exhibits better performance than other language models. It is evident that fine-tuning the hyperparameters plays a crucial role in the model's performance, as the results show a difference of up to 6.5% in the F1-score. This highlights the need for proper tuning to achieve optimal results and underscores the significance of this aspect in the development of language models.

Figure 4 reports different results obtained by evaluating each model in various configurations on the test set. We noticed high variability in performance based on the model's hyperparameters.

Overall, our approach has been shown to perform comparably with the benchmark baseline models while exhibiting slightly improved results in terms of recall. We believe that this performance is the result of the specific data-sampling technique that we implemented during the training phase, which aimed to balance the data's distribution. By leveraging this approach, we were able to address the class imbalance and improve the models' performances effectively.

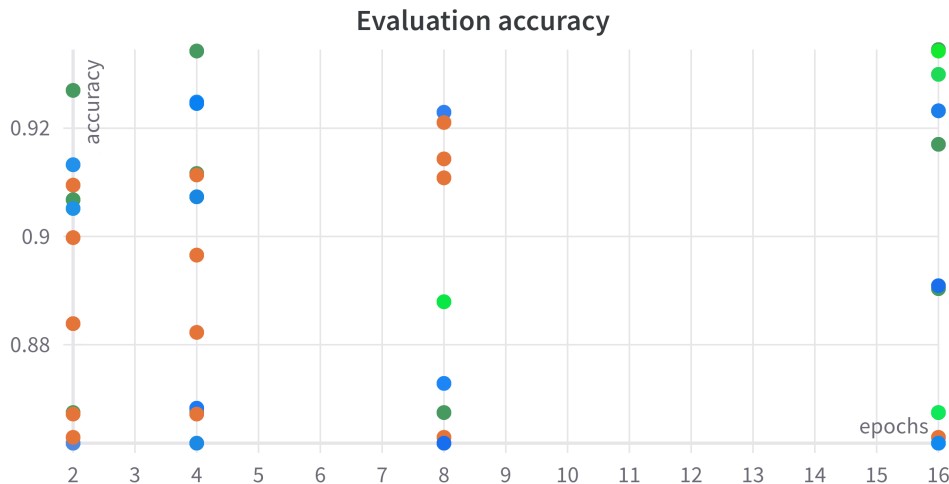

**Figure 4.** Our different models' prediction accuracies. The green, blue, and orange circles represent BioBERTa, BioBERT, and BERT models respectively

### 4.3. Interpretability of the Generated Text

The interpretability of models, as illustrated in Figure 5, plays a crucial role in understanding a model's decision-making process and predictions, especially in medical applications [37]. Using fuzzy theory in the defuzzification processes helps to deal with uncertain and ambiguous information. Still, this uncertainty can also impact the interpretability of the models trained on such data.

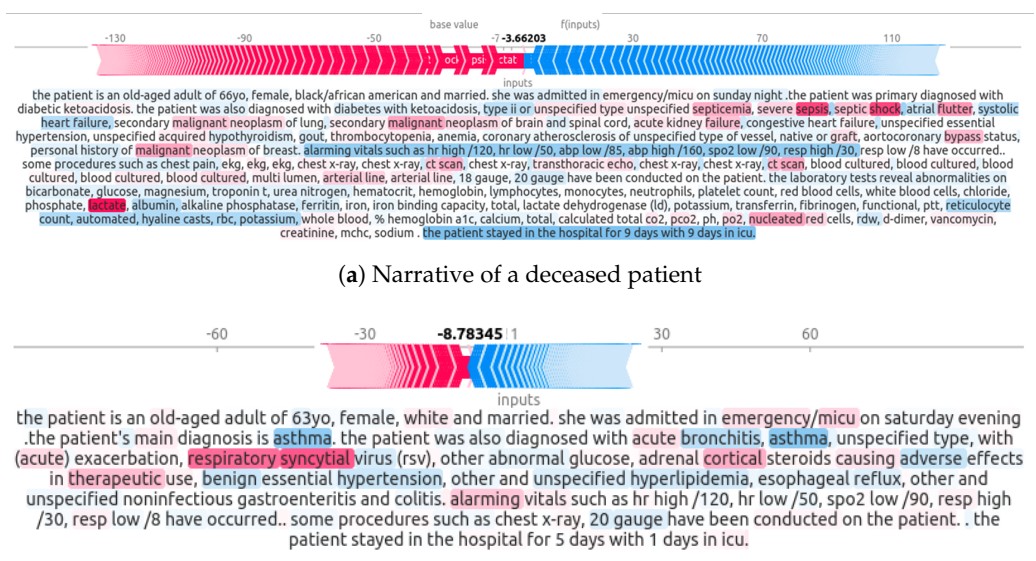

(**a**) Narrative of a deceased patient

(**b**) Narrative of a discharged patient

**Figure 5.** Interpretability visualization using shapley additive explanations (SHAP) on the narratives from two different classes.

The explainability of a model based on the text generated from a defuzzification process depends on various factors, such as the choice of the defuzzification method, the structure of the model, and the complexity of the generated text. Rule-based text provides more nuanced data by structuring the narratives into a more comprehensive and interpretable construction.

Figure 5 shows a visualization of BioBERTa of two generated texts using Shapley values [38], revealing the importance of each token. Red regions correspond to parts of the

text that increase the model's output when they are included, pushing the model to predict the patient as "deceased". In contrast, blue regions decrease the output of the model to predict a "discharged" patient.

It can be seen in Figure 5a that even if vitals and length of stay helped the model to increase the output values, the shade of red seen on "*sepsis*" and "*lactate*" was too dark to predict the fatal outcome. This is more understandable, as sepsis is a life-threatening on its own; a high serum lactate level as a consequence of sepsis may predict death within 24 h [39]. Figure 5b shows a narrative where the primary diagnosis indicates a significant contribution to the survival of the patient, despite an elevated value being given to the patient's coinfection with the "*respiratory syncytial virus*", a less lethal infection [40].

## 5. Conclusions

This research aimed to analyze the impact of the generated data on the prediction of in-hospital outcomes. The defuzzification process of generating narratives involved extracting features from the MIMIC-III dataset and fusing them to represent the patients exhaustively. The generated data were then subjected to a grammatical assessment to eliminate errors and improve the quality of the generated narratives. The data were generated for 37,110 admission IDs in the dataset, and the lengths of the narratives varied based on the number of parameters each patient had.

To train and evaluate the models, 10,116 input sentences were used, and the performance was tested on 2529 narratives. The BERT, BioBERT, and BioBERTa models were trained using the BERT-based, uncased tokenizer and the BioBERT tokenizer, respectively. The study also involved hyperparameter optimization, where a random search algorithm was used to select the optimal values of hyperparameters, such as the batch size, learning rate, and training epochs. The best performance was achieved with a batch size of 32, 4 training epochs, and a learning rate of $3.86 \times 10^{-5}$.

The evaluation of the models was based on the prediction of the outcome of the patient's hospitalization, where the data were labeled as zero for patients who were discharged and one for those who died. The results were measured using the F1-score, precision, and recall for the fatality and survivor classes. The results demonstrated the highly competitive performances of the BERT and BioBERT models. BioBERTa exhibited better performance compared to the other language models. The results showed that the best performance was achieved using a maximum input length of 512 tokens, with optimized hyperparameters.

In conclusion, this study demonstrates that FL- and rule-based approaches can play a significant role in generating comprehensive and interpretable medical narratives to describe a patient extensively. The results of the study show the potential of fine-tuned language models such as BioBERTa for improving the accuracy of predictions and provide a better understanding of the hospitalization outcomes of patients. The interpretability of models trained on the text generated from a defuzzification process is crucial for ensuring the transparency and reliability of the model's predictions.

However, our approach has two significant limitations. Firstly, the accuracy of the resulting model depends directly on the size of the universe of discourse grouping the classes, and for some features, there is no deterministic way of establishing boundaries between classes. Subsequently, this approach requires domain expertise to determine the appropriate linguistic rules, and there is potential for bias in the textualization process. Additionally, the experimental results show that the performance of LM relies heavily on hyperparameter fine-tuning.

In future work, we plan to explore the use of neuro-fuzzy theory, in combination with current state-of-the-art LMs, and investigate methods for reducing expert dependency by incorporating external data sources, such as ontology. Overall, this study provides a step toward improving healthcare outcomes through data-driven decision-making processes.

**Author Contributions:** Conceptualization, C.M. and I.P.; Software, C.M.; Formal analysis, I.P.; Data curation, C.M.; Writing—original draft, C.M.; Writing—review & editing, I.P.; Visualization, C.M.; Supervision, I.P.; Project administration, I.P. All authors have read and agreed to the published version of the manuscript.

**Funding:** This research received no external funding.

**Institutional Review Board Statement:** This research was conducted on MIMIC3 dataset which requires a Collaborative Institutional Training Initiative(CITI) certification for Data or Specimens Only Research. The data access was credentialed by the Record ID: 53185588 under the first author's name which can be verified at https://www.citiprogram.org/verify/?k807e5a87-60b4-4aad-9dd8-3 f2ba5366b69-53185588 (accessed on 27 January 2023).

**Data Availability Statement:** MIMIC III datasets on PhysioNet is a credentialed-only access. It is accessible by registered users who complete the credentialing process and sign a Data Use Agreement. Public data sharing is not allowed. Specific training requirements can be found at https://physionet. org/about/database/#credentialed (accessed on 27 January 2023).

**Conflicts of Interest:** The authors declare no conflict of interest.

## Abbreviations

The following abbreviations are used in this manuscript:

| | |
|---|---|
| BERT | Bidirectional Encoder Representations from Transformers |
| FL | Fuzzy Logic |
| MIMIC | Medical Information Mart for Intensive Care |
| LM | Language Model |
| NLP | Natural Language Processing |

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
