# Peer review of "Bridging the Gap between Medical Tabular Data and NLP Predictive Models: A Fuzzy-Logic-Based Textualization Approach"

_electronics, doi:10.3390/electronics12081848_

Round 1
Reviewer 1 Report
see attachment

Author Response
Thank you for your time and consideration of our paper. We sincerely appreciate your feedback and criticism toward improving the quality of our work. Please find below a detailed description of our steps to address your highlighted issues:
- “Authors claims that F1-score is 93.7%. have authors compered these data?”
We calculated the accuracy of our NLP-based predictive models using F1-score, among other metrics. We utilized those metrics mainly because they were used in the baseline research cited with ref. 11. This reported accuracy relates to the ability of the model to distinguish two different classes of the outcomes present in the dataset, which are dead or discharged.
- “Authors have tried to identify the gaps, but it was not written in proper manner. I emphasized that authors should clearly mention the literature gaps and novelty of the proposed work.”
Thank you very much for your suggestion. We have extensively revised our manuscript in consideration of your recommendation. Please, check the new provided manuscript.
- “Sentence “Traditionally, fuzzy logic is a science that makes machines think and understand the way humans do [16] by proposing fuzzy sets to manage imprecise and vague knowledge [17].” The way of writing the reference in the mid of sentence is not good practice. Pls make sure all this type of the references should not be there.”
Thank you for highlighting this mistake, we fixed it in the revised version, and we also made sure to check everywhere else that may have occurred.
- “Pls check eq 6, there is some mistake.”
We appreciate very much your attention to detail. Our understanding is that the equation calculates the MOM by first identifying the maximum value of a certain function, $\mu A(x_i)$, among all the values in the dataset. Then, it calculates the arithmetic mean of all the values in the dataset that have this maximum value. The numerator of the equation sums the product of each value in the dataset, $x_i$, with a Boolean indicator function that evaluates to 1 if $\mu A(x_i)$ equals the maximum value and 0 otherwise. The denominator sums the same Boolean indicator function over all the values in the dataset. Dividing the numerator by the denominator gives the MOM. In other context, this equation could be simplified as “$ y = \frac{1}{N} \sum_{i=1}^N \max{\mu_A(x_i)}$” as stated in this reference.
If our understanding is wrong, we would like to ask for your clarification on the mistake mentioned.
- “Line 191, “We selected patients admitted or transferred to the Cardiac Surgery 191 Recovery Unit (CSRU), Medical ICU(MICU), Surgical ICU (SICU), and “emergency or 192 Urgent” as ADMISSION_TYPE.” Don’t use “we” in the manuscript.”
Thank you for your recommendation. We addressed this issue by rewriting the paragraph more comprehensively to explain this critical step in data extraction.
- “Caption of the y ordinate in figure 2 should be corrected.”
Thank you again for your suggestion. However, we couldn’t understand the reason why you recommended correcting the caption. We will be happy to fix it if you let us know what is wrong with it.
- “Why authors have classified blood pressure in only four group “”Normal”, ”Elevated”, ”High”, and ”Hypertensive crisis””
Thank you for this question. This reference was a simple example to illustrate a fundamental solution to the problem of clustering terms to represent ranges of numerical values. However, our practical approach used prior published work to divide clinical values into classes, as reported in Table 1. We understand that constructing our fuzzy sets has some limitations since there is no deterministic way of establishing boundaries between classes. One of the consequences mentioned in the paper is a direct dependency of the resulting model on the size of the universe of discourse. This matter should be addressed and mentioned in future work.
- In conclusion, the should be some quantitative results.
We have added more recent analysis results to support more accurately the mentioned results. Figure 4 also shows the variation of models of different configurations.
Reviewer 2 Report
In this paper, authors proposed a Fuzzy logic-based pipeline that generates medical narratives from structured data. They evaluated this rule-based approach in predicting patient outcomes. The pipeline includes a feature selection operation, a reasoning and inference function, and a textualization process that addresses the uncertainty and vagueness inherent in medical data. They then perform an extensive evaluation of the generated narratives using transformers-based NLP models on a patient outcome prediction task. They furthermore assessed the interpretability of the generated text using Shapley values. It is a well-structured paper with interesting results. However, it requires further improvements.
(1) In Section 1, the main contributions of this paper should be further summarized and clearly demonstrated. This reviewer suggests the authors exactly mention what is new compared with existing approaches.
(2) The research gaps in the abstract and introduction should be clearly expressed. Please rewrite this part.
(3) The authors must clearly explain the difference(s) between the proposed method and similar works in the introduction. The authors should further highlight the manuscript's innovations and contributions.
(4) In Line 326, "Using BERT tokenizer, the lengths were 1686, 67 tokens for the longest and shortest sentence with a median of 258". How to determine these values? Please provide the detailed describings.
(5) In page 13, Figure 3 is not clear. Please revise it.
(6) The inspiration of your work must be highlighted. For example, 10.1109/TII.2022.3232766; 10.1016/j.marstruc.2022.103338; 10.1016/j.ins.2022.11.019; 10.3389/fendo.2022.1057089 and so on.
(7) There are a few typos and grammar errors in the manuscript. Please polish the manuscript carefully.
Author Response
Thank you for your time and consideration of our paper. We sincerely appreciate your feedback and criticism toward improving the quality of our work. Please find below a detailed description of our steps to address your highlighted issues:
- “In Section 1, the main contributions of this paper should be further summarized and clearly demonstrated. This reviewer suggests the authors exactly mention what is new compared with existing approaches.”
Thank you so much for your feedback. We have extensively revised section 1 to ensure our contribution is clear to the reader.
- “The research gaps in the abstract and introduction should be clearly expressed. Please rewrite this part.”
This concern has been addressed together with the previous one mentioned above. We revised the introduction, the literature review, and the abstract.
- “The authors must clearly explain the difference(s) between the proposed method and similar works in the introduction. The authors should further highlight the manuscript's innovations and contributions.”
We deeply revised the introduction and the background section to highlight the main difference between our approach and the existing work, as well as the contribution of our approach.
- In Line 326, "Using BERT tokenizer, the lengths were 1686, 67 tokens for the longest and shortest sentence with a median of 258". How to determine these values? Please provide the detailed describings”
Thank you for raising this question. Using the BERT tokenizer, we counted the tokens of each input sentence, and the results were 1686 and 67, respectively, for the longest and shortest sentence with a median of 258 tokens. We reformulated the sentence in the manuscript to make it clear to the reader.
- “In page 13, Figure 3 is not clear. Please revise it.”
We couldn’t figure out which figure is mentioned in this feedback. On page 13, we have figure 4, while figure 3 is on page 11.
However, for more clarification, figure 13 shows the training process results while searching for the best combination of hyperparameters. This combination involves a lot of hyperparameters and models that couldn’t be reported as a caption for this figure. Nevertheless, we will annotate the axis for the figure interpretation.
- “The inspiration of your work must be highlighted. For example, 10.1109/TII.2022.3232766; 10.1016/j.marstruc.2022.103338; 10.1016/j.ins.2022.11.019; 10.3389/fendo.2022.1057089 and so on.”
We carefully surveyed the suggested references. However, unless there’s a mistake in the reference, we couldn’t identify an inspirational relationship between our work and the following papers(“10.1109/TII.2022.3232766”, “10.1016/j.ins.2022.11.019” , “10.1016/j.marstruc.2022.103338;”). In case there was a mistake, please provide us with the correct references.
- “(7) There are a few typos and grammar errors in the manuscript. Please polish the manuscript carefully.”
Thank you for your review and for bringing the typos and grammar errors to our attention. We apologize for any inconvenience this may have caused. We carefully reviewed and polished the manuscript to ensure it meets the highest quality standards. We appreciate your feedback and look forward to delivering an improved version of the manuscript.
Reviewer 3 Report
Please revise the paper in the following parts:
1) Introduction is too short to serve the purpose. It needs to be extended further.
2) Please refine the key contributions of the paper with more details (in terms of originality of the work).
3) Please include a summary about the paper organization at the end of Section 1.
4) Please specify key results and research insights in a subsection of Section 4.
5) Conclusions need to be re-organized. Please move its first 3 paragraphs along with the Figure 4 to Results and Discussion Section.
6) Please supply limitations of the work at the end of the Conclusions.
Author Response
Thank you for your time and consideration of our paper. We sincerely appreciate your feedback and criticism toward improving the quality of our work. Please find below a detailed description of our steps to address your highlighted issues:
- “Introduction is too short to serve the purpose. It needs to be extended further.”
We extensively revised the abstract to give it the consistency that represents the work presented in this paper.
- “Please refine the key contributions of the paper with more details (in terms of originality of the work).”
We revised the introduction section to redefine the key contribution of our paper, as well as highlight the novelty of the approach. - "Please include a summary about the paper organization at the end of Section 1"
Thank you for your valuable feedback and suggestion. We appreciate your input and have carefully considered your request. We incorporated a summary of the paper organization at the end of Section 1 to provide readers with a clear understanding of the paper's overall structure.
- “Please specify key results and research insights in a subsection of Section 4.”
The key results of our paper have been provided and discussed in each subsection of Section 4.
Our choice was guided by the research questions listed in Section 1. We dedicated a subsection to present the key finding as well as the results individually
- “Conclusions need to be re-organized. Please move its first 3 paragraphs along with the Figure 4 to Results and Discussion Section.”
We addressed this issue by revising the conclusion and the results sections. We noticed that Figure 5 automatically moved to the conclusion section while it was declared in the results section.
- “Please supply limitations of the work at the end of the Conclusions.”
We revised this section to include a comprehensive paragraph on the limitations of our research.
Reviewer 4 Report
In this paper, we are proposing a Fuzzy logic-based pipeline that generates medical narratives from structured data.
The study is interesting and the selected papers are good.
However, the paper needs to acquire more quality in terms of cited papers.
For this purpose I suggest to include the following paper among the cited papers because, in my opinion, it is crucial introducing some of model checking papers:
"Heuristic strategies for assessing wireless sensor network resiliency: an event-based formal approach". Journal of Heuristics 21, 145-175
I am confident if the authors add the citation and they motivate the importance to consider it, the paper will acquire more quality for the publication.
Author Response
Thank you for your time and consideration of our paper. We sincerely appreciate your feedback and criticism toward improving the quality of our work. Please find below a detailed description of our steps to address your highlighted issues:
“..., However, the paper needs to acquire more quality in terms of cited papers. For this purpose I suggest to include the following paper among the cited papers because, in my opinion, it is crucial introducing some of model checking papers:
"Heuristic strategies for assessing wireless sensor network resiliency: an event-based formal approach". Journal of Heuristics 21, 145-175
I am confident if the authors add the citation and they motivate the importance to consider it, the paper will acquire more quality for the publication.”
Dear reviewer, we are grateful for your time reviewing our paper. Your kind feedback and effort to improve our work are valuable.
We’ve carefully surveyed the suggested reference. However, unless there’s a mistake in the reference, we couldn’t identify an inspirational relationship between our work and the recommended paper. In case of a mistake, we would appreciate it if you could provide us with the correct reference.
Reviewer 5 Report
The manuscript gives another example of classic data processing, already widely applied to medical processes. It shows how linguistic schemes help to enter health data into a system that supports automated reasoning. The text is conventionally structured, but the grammar is not always regular English because of stiffly composed sentences. Overall, the manuscript is properly put together.
The title of the proposed paper uses all the relevant keywords for the research area but makes not much sense in its combination. What is the problem to be solved or the process to be analyzed? In other words: what will the attention of a reader be drawn? The doctor takes measurements, reads values and draws conclusions. Documents are only required for communication beyond the doctor/patient relation?! It is only later in the text that such background ideas become apparent.
For me, the paper just started at section 3. It was only here that the authors reveal, that they are not answering the question of the practitioners in the field on the language-independent communication with patients _ a search that involved many people and produced many medical products over the past decades _, but demonstrating how these same tools can be used to mine Big Health Data. It would have been helpful for the older readers (like me) if the different search question has been sharply demarcated from the start.
The research questions are formulated before their target becomes apparent in section 3. Actually, they are not presented in questions but in a listing of sections to come. Consequently, the answers in section 4 are vague and the claim of an innovative approach lacks substantiation. It is merely showing how the results from the database can be extracted as natural text, but this technology is similar to commerce products that came into use around 2000 to generate for instance job advertisements from daily journals.
The literature references are not always complete or appropriate. For instance, reports should provide the publishing organization, Internet links should have a time stamp and preferably be given in the page header, and magazine / book locations must give the complete page location. The reference should clearly show what kind of knowledge carrier is used.
For the work presented in this manuscript, I suggest to apply also the neuro-fuzzy work as published by Jang, Sun and Mizutani in 1997. It was available as part of the MatLab software. In terms of the subsection mentioned in subsection 3.4, this technology bridges more easily the data gaps, typical for intelligent harvesting of large-scale health data.
My advice to the authors is to re-consider the aims and scope. Then they can re-order the material and a much more targeted article will emerge. This looks much more work than it will actually be.
Author Response
Thank you for your time and consideration of our paper. We sincerely appreciate your feedback and criticism toward improving the quality of our work. Please find below a detailed description of our steps to address your highlighted issues:
- “The title of the proposed paper uses all the relevant keywords for the research area but makes not much sense in its combination. What is the problem to be solved or the process to be analyzed? In other words: what will the attention of a reader be drawn? The doctor takes measurements, reads values and draws conclusions. Documents are only required for communication beyond the doctor/patient relation?! It is only later in the text that such background ideas become apparent.”
Thank you so much for taking the time to review our work deeply. We revised the title to suit the content of the paper better. We hope this new title describes the work presented in it well.
- “For me, the paper just started at section 3. It was only here that the authors reveal, that they are not answering the question of the practitioners in the field on the language-independent communication with patients _ a search that involved many people and produced many medical products over the past decades _, but demonstrating how these same tools can be used to mine Big Health Data. It would have been helpful for the older readers (like me) if the different search question has been sharply demarcated from the start.”
We extensively revised the paper to highlight the prior related work as well as connect the introduction to the rest of the paper. We hope that our revision helps to understand the research motivation and question.
- “The research questions are formulated before their target becomes apparent in section 3. Actually, they are not presented in questions but in a listing of sections to come. Consequently, the answers in section 4 are vague and the claim of an innovative approach lacks substantiation. It is merely showing how the results from the database can be extracted as natural text, but this technology is similar to commerce products that came into use around 2000 to generate for instance job advertisements from daily journals.”
Thank you for your valuable remark. We carefully considered all your remarkable feedback to improve the connection between sections taking into account the writing methodologies. We hope that our revision will make the paper coherent. In addition, we highlighted the novelty of our approach regarding its application in the medical area and its utility in patient outcome prediction using modern language models. - "The literature references are not always complete or appropriate. For instance, reports should provide the publishing organization, Internet links should have a time stamp and preferably be given in the page header, and magazine / book locations must give the complete page location. The reference should clearly show what kind of knowledge carrier is used."
Thank you for your feedback regarding the literature references in our manuscript. We carefully reviewed and revised the references to ensure that they are complete and appropriate. Specifically, we included a time stamp for Internet links and provided complete page locations for book references. Your feedback has been valuable in improving the quality of our manuscript, and we thank you for your thorough review. - “For the work presented in this manuscript, I suggest to apply also the neuro-fuzzy work as published by Jang, Sun and Mizutani in 1997. It was available as part of the MatLab software. In terms of the subsection mentioned in subsection 3.4, this technology bridges more easily the data gaps, typical for intelligent harvesting of large-scale health data.”
Thank you very much for this thoughtful suggestion. Neuro-fuzzy systems, which combine the learning ability of neural networks with the reasoning ability of fuzzy logic, can allow for the creation of models that can handle incomplete and uncertain data. These models can then be used to predict outcomes and make decisions based on the available data more automatically. We believe that this pipeline has the downside of limiting the control over the predictive models.
In the context of our paper, the fuzzy logic-based textualization approach can be seen as a means of converting tabular data into natural language text that can be better understood and analyzed by NLP predictive models. The use of fuzzy logic allows for handling imprecise and uncertain data in this conversion process, similar to how it can be used in neuro-fuzzy systems for predictive modeling. This way, having a clear separation of data extraction and modeling gives our approach more flexibility to control inputs for a desired output. Overall, our work follows the footsteps of the research described in the suggested book by emphasizing the importance of addressing uncertainty and interpretability. In addition, we understand that the book is an excellent reference for our work, and we should have mentioned it as a main reference for several reasons. We considerably revised the literature by emphasizing the relationship/difference between these two works. Moreover, we understand that there must be a separate research to deeply study and compare these two approaches to support our theories. Hence, this has been mentioned as an alternative future work.
Round 2
Reviewer 2 Report
This paper can be accepted now.
Author Response
Dear Reviewer,
I wanted to take a moment to express my sincere gratitude for your valuable feedback on my paper. Your expert opinion and insightful comments have been tremendously helpful in improving the quality of my work. I am delighted to hear that you have suggested my paper for acceptance and your recommendation means a lot to me.
Once again, thank you so much for your time and effort in reviewing my paper. I truly appreciate your contribution to the advancement of my research field.
Reviewer 4 Report
The paper could be published. Everything is ok.
Author Response
Dear Reviewer,
I am writing to express my sincere gratitude for your invaluable feedback on my paper. Your expert insights and comments have greatly contributed to enhancing the quality of my work. Your recommendation for acceptance is particularly appreciated and means a lot to me.
I am grateful for the time and effort you dedicated to reviewing my paper and for your contribution to the advancement of my research field.
Thank you again for your support.
Reviewer 5 Report
The manuscript tells on a topic in classic data processing, already widely applied to medical processes. In this area, linguistic schemes help to enter health data into a system that supports automated reasoning and the results can be extracted for further inspection. Though the title does not give a clue, it seems that the goal is to generate the output in prose. Overall, the manuscript is properly put together. But the relevance of the research is not clarified. The subject is the evaluation of the processing pipeline earlier proposed by the authors but it remains unclear what the purpose is. The introduction leads not to a problem statement nor to a research identification but only to a work listing.
The authors base their research subject on another paper that loosely mentions a potential need for their technology. The subject is not necessarily answering the question on the language-independent communication with patients posed by practitioners in the field _ a search that involved many people and produced many medical products over the past decades _. It just serves to demonstrate how these same tools can be used to mine Big Health Data. But it appears that the work has already been done and published in IEEE Access, pgs. 16489 – 16498.
Where is the innovation? Section 2 gives textbook information on the potentially useful technologies. Section 3 gives the research as previously published by the authors. Then, in section 4, close to the end of the paper, some experimentation is discussed to substantiate the claim of an innovative approach. It is merely showing how the results from the database can be extracted as natural text, but this technology is similar to commercial products that came into use around 2000 to generate for instance job advertisements from daily journals. To substantiate the authors’ claim, it is reasonable to hear what they improve on the EHR software that started to enter the market place around 2014.
The literature references are not always complete or appropriate. For instance, reports should provide the publishing organization, Internet links should have a time stamp and preferably be given in the page header/footer, and magazine / book locations must give the complete page location. The reference should clearly show what kind of knowledge carrier is used. The newly added or changed text has more grammatical errors than before. Furthermore, the earlier work of the authors is not mentioned or (as lit. 7 and lit. 35) duplicated.
The manuscript is clearly the result of ongoing work of the authors over the past 4 years. The most recent work is duplicated and not cited. Especially it is needed to have clear demarcation lines between the distinct publications: what is the original material and, if so, where is the origin. Especially the data origin must be pointed out in detail, and not just to the volume, as this serves repeat the calculations.
Round 3
Reviewer 5 Report
The third version has not improved the document. On the contrary, some cosmetic details have been made. The similar publications of the same authors are still not placed in perspective. The back-end neural circuit is numerically dubious, and should not be included.